# Effects of Nutritional and Social Factors on Favorable Fetal Growth Conditions Using Structural Equation Modeling

**DOI:** 10.3390/nu14214642

**Published:** 2022-11-03

**Authors:** Gugulethu Moyo, Zachary Stickley, Todd Little, John Dawson, Shera Thomas-Jackson, Jennifer Ngounda, Marizeth Jordaan, Liska Robb, Corinna Walsh, Wilna Oldewage-Theron

**Affiliations:** 1Department of Nutritional Sciences, Texas Tech University, Lubbock, TX 79409, USA; 2Center for Health and Wellbeing, Princeton School of Public and International Affairs, Princeton University, Princeton, NJ 08544, USA; 3Institute for Measurement, Methodology, Analysis, and Policy, Texas Tech University, Lubbock, TX 79409, USA; 4Department of Economics, Applied Statistics, and International Business, New Mexico State University, Las Cruces, NM 88003, USA; 5Department of Human Development and Family Sciences, Texas Tech University, Lubbock, TX 79409, USA; 6Department of Nutrition and Dietetics, University of the Free State, Bloemfontein 9300, South Africa; 7Department of Sustainable Food Systems and Development, University of the Free State, Bloemfontein 9300, South Africa

**Keywords:** maternal, diet, health, stress, social support, birth outcomes, HIV, GBMI, food security

## Abstract

Poor birth outcomes such as low birth weight, low birth length and short gestational age, are public health concern issues in South Africa (SA). This study utilized structural equation modeling (SEM) to explore how nutritional and social factors contribute to favorable fetal growth conditions (FFGC) in pregnant women living with and without human immunodeficiency virus (HIV), in the Free State Province of SA. Sociodemographic characteristics, stress, health and nutrition-related information, and birth outcomes data were collected and analyzed from a subsample of 305 women enrolled in a cohort study from 2018–2020. Descriptive statistics were analyzed in R version 4.1.2 and SEM was conducted in Lavaan version 0.6–5. Higher gestational body mass index (GBMI) and income levels were associated with higher FFGC (*p* < 0.05). Household incomes were positively associated with dietary micronutrient quality (*p* = 0.002), GBMI (*p* = 0.012) and food security (*p* = 0.001). Low incomes (*p* = 0.004) and food insecurity (*p* < 0.001) were associated with higher stress, while social support was positively associated with food security status (*p* = 0.008). These findings highlight the complex interconnections between the social and nutritional factors that are associated with fetal growth conditions. Multisectoral community-based programs may be a useful strategy to address these challenges.

## 1. Introduction

Annually, about 15 million infants are born preterm (<37 weeks of gestation), mostly in Sub-Saharan Africa and South Asia [1]. Preterm births contribute to over one third of neonatal deaths and are associated with 16% of the deaths occurring in children under five [2]. Additionally, over 20.5 million infants globally have a low birth weight (below or equal to 2500 g) [3]. Low birth weight is a sign of fetal growth restriction and is associated with 80% of neonatal deaths [4]. Apart from the increased risk of mortality, poor birth outcomes have also been associated with increased risk for non-communicable diseases in later life as supported by The Developmental Origins of Health and Disease (DOHaD) hypothesis. The DOHaD was developed by David Barker, based on the results of observational studies that suggested that adverse nutritional and environmental exposures in the womb can have a negative impact on the adult life of the exposed fetus [5,6,7]. Examples include an increased risk of metabolic syndrome, diabetes, hypertension, and cardiovascular disease prevalence in adulthood [8,9,10,11,12,13].

The World Health Organization (WHO) has identified poor birth outcomes as a key indicator of a “multifaceted public-health problem” [14]. One of the key global strategies to address this is the 1000 Days Approach [15]. According to the 1000 Days Approach, the time from conception to when a child is two years of age, the first 1000 days of life, is considered “a critical window of opportunity for laying down the foundations for optimum health and development” [15]. During pregnancy, various challenges affecting the pregnant woman can directly or indirectly affect fetal growth conditions. These include micronutrient deficiencies, inadequate dietary intake, alcohol and drug abuse, poverty, underweight or obesity during pregnancy, trauma, and high stress levels [16,17]. Measuring fetal growth conditions is challenging since this is a relatively latent construct that cannot be measured directly. Past studies have largely relied on low birth weight as a proxy indicator for the fetal environment [18]. However, Bollen, and colleagues, developed a latent variable called Favorable Fetal Growth Conditions (FFGC), based on the manifest variables of birth length, birth weight and gestational age. This FFGC latent variable provides a more comprehensive estimate of the fetal environment, compared to relying on only birth weight [19]. 

In South Africa (SA), as many as 14.2% infants had a low birth weight in 2016 [20], and relatively high prevalence of preterm births at 15–17% in 2019 [21]. This is despite a comprehensive health care sector, offering a wide range of preventive and curative services to the population [22,23,24,25]. There has been relatively little progress made in reducing the prevalence of low birth weight and preterm births [26] and this may be due to the multifactorial contributors to FFGC. There is a need to design evidence-based solutions and strategies that address the relatively high prevalence of poor birth outcomes in SA, but currently the social and nutritional factors potentially contributing to FFCG through complex interconnected pathways are poorly understood [27]. There are currently no known studies that have used a holistic approach to explore these within the SA context. Thus, the present study utilized structural equation modeling to address this gap by investigating the social and nutritional determinants of FFGC in pregnant women living in the Free State Province, SA. 

## 2. Materials and Methods

This research was conducted in the Free State province of SA, through a collaboration between researchers in the United States of America (USA) and SA. Due to the relatively high prevalence of human immunodeficiency virus (HIV) in SA (19%) [28], a multigroup analysis was conducted to assess model fit in women living with and without HIV. 

### 2.1. Study Design and Participants

A cohort study design was utilized, and women were recruited from a regional public hospital in Bloemfontein. All pregnant women (approximately 700 per month) attending the antenatal clinic at this hospital from May 2018–April 2019 were eligible to participate in the study if they met the inclusion criteria that included 18 years or older, at 12 weeks gestation or more, able to speak English, Afrikaans, or Sesotho and had provided written informed consent to participate in the study after receiving an explanation and documentation related to the study and its procedures. A total of 682 women enrolled in the study. We confined our analysis to information to a subsample of 305 women who completed the study and provided full information related to birth outcomes and HIV status. Figure 1 summarizes the participants and attrition throughout the study.

### 2.2. Data Collection

Data were collected from May 2018 to April 2019 at a regional hospital’s antenatal clinic using structured interviews administered by trained fieldworkers in the participant’s language of choice (English, Sesotho, or Afrikaans). 

Sociodemographic data was obtained using the Socio-demographic and Household Questionnaire that collected information related to age, marital status, education, employment, and income status, household size, facilities, and assets. Information related to stress and social support was determined from the Health, Lifestyle and Pregnancy History questionnaire [29] that collected information related to exposure to crime, violence, debt, breakups, grief, caring for someone seriously ill, alcohol use and social support. 

Anthropometric measurements (heights and weights) were collected using the standardized techniques specified by the International Society for the Advancement of Kinanthropometry (ISAK) [30]. Additional information related to pregnancy stage (weeks), and pre-pregnancy weight was also collected to allow for the computation of gestational body mass index (GBMI) [31]. The GBMI was then classified as underweight (≥10 to <19.8 kg/m²), normal weight (≥19.8 to <26.1 kg/m²), overweight (≥26.1 to <29 kg/m²), or obese (≥29 kg/m²) [32]. 

A quantified food frequency questionnaire [33] was used to estimate dietary intake. Food models, household measures and photographs were used to aid in the estimation of portion sizes. Foods were converted to total grams consumed in the past 28 days using the South African Medical Research Council (SAMRC) Food Quantities Manual [34] and the Condensed Food Composition Tables for South Africa [35], and daily nutrient intakes were determined. The daily nutrient intake data was used to calculate nutrient adequacy ratio (NAR) and mean adequacy ratio (MAR) for each participant [36,37]. 

Household food security was assessed through the Household Food Insecurity Access Scale (HFIAS) [38] that measured food security based on a recall period of 28 days. Households were categorized as food secure, mildly food insecure, moderately food insecure, and severely food insecure based on the scoring of certain questions [39]. HFIAS responses were also converted into a continuous indicator that ranged from 0 to 27, indicating the degree of insecure food access. The higher the score, the more food insecure the participant [38].

Women were contacted via text message after their expected delivery date to remind them to provide follow-up information related to their birth outcome. Neonate information relating to gestational age at delivery, anthropometric measurements, congenital disabilities, and HIV exposure was obtained from the Road to Health Booklet. The Road to Health book is provided for all new births by health facilities [39] and is completed by trained healthcare staff at each hospital or clinic visit. 

### 2.3. Development of Latent Variables

There were three latent variables included in the model, namely two independent variables, stress, and dietary micronutrient quality (DMQ) and the outcome variable, FFGC. Stress was determined from the Health, Lifestyle and Pregnancy History questionnaire that collected information regarding exposure to crime and violence, debt, divorce and breakups, grief or caring with a seriously ill loved one [29]. DMQ was based on nine nutrients Vitamin A, B6, B12, C and D, folate, iodine, iron, and zinc. The importance of these nutrients during pregnancy has been described by the WHO and recent studies [40,41,42]. To create the indicators for stress and DMQ, a technique called parceling was used (where two or more indicators are averaged to create a new variable which is then used as the manifest indicator of the latent construct under study) [43]. At times a manifest variable may only tell a part of the story, thus combining several variables into a parcel creates a more reliable indicator. Parceling is a technique that improves the quality of the indicator, and averaging, as opposed to summing the items allows the parcels to have similar metrics [43]. Based on the balancing approach, items with the strongest item–scale correlations were paired with those with the weakest, and this process was repeated until all items were parceled. Favorable Fetal Growth Conditions is an abstract variable developed by a group of statisticians [19], which seeks to explain the combined effects of environmental, genetic and health related factors on the development of the fetus in-utero. It is these in-utero exposures that are said to influence the future adult health outcomes of the fetus, according to Barker’s DOHaD hypothesis [44]. 

### 2.4. Data Analysis

Descriptive statistics were calculated in R statistical package version 3.6.1, while Lavaan version 0.6–5 in R version 3.6.1 was used to create a structural equation model (SEM). Structural equation modeling was used to model the relationships between stress, social support, food security status, household income, GBMI, dietary micronutrient quality, and FFGC in pregnant women living with (n = 104) and without (n = 201) HIV in the sample. Due to the relatively high prevalence of HIV in SA (19%) [28], a multigroup analysis was conducted to compare outcomes in women living with and without HIV. In terms of adequacy of the sample size, based on a series of Monte Carlo simulations, a sample size of 100 participants for single group models and 75 per group for multigroup models has been estimated as large enough to be adequately powered for SEM [43]. Based on this assertion, it was adequately powered for the multigroup model analysis conducted. The data from women who provided HIV-related information (n = 305) was utilized to estimate the model using the full information maximum likelihood (FIML), to estimate missing data. A *p*-value less than 0.05 (*p* < 0.05) was considered statistically significant.

### 2.5. Hypothesized Diagram

The path diagram (Figure 2) summarizes the hypotheses. The single headed arrows pointing towards the FFGC represent the regression associations being hypothesized, while the double headed arrows connecting stress, social support, household income, GBMI, household food security and DMQ, represent the covariance associations that were hypothesized. The measurement models for the latent constructs are reflective (with arrows pointing away from circles which represent the latent construct). 

## 3. Results

### 3.1. Sociodemographic Characteristics

Women living with HIV were slightly older at enrollment than those without HIV (*p* = 0.014), but there was no significant difference in gestational age at enrollment, as summarized in Table 1.

The majority of women were married or in a relationship, but significantly more women without HIV were married (*p* = 0.0094) or in a relationship (*p* = 0.021), when compared to women living with HIV. The majority of women in both groups completed secondary level education, however, a significantly higher proportion of women without HIV had completed tertiary level education (*p* = 0.024) compared to women living with HIV, and a similar trend was observed for their partners’ education. Both groups had high levels of social support, with over 80% of all participants reporting that they had several people to turn to for social support. There were no differences between the stressors experienced by the two groups, except significantly more women with HIV had a close family member who was seriously ill (*p* = 0.014), compared to women without HIV.

### 3.2. Confirmatory Factor Analysis

In this stage the relationship between the latent variable and its indicators was assessed. Parceled indicators were used for stress and DMQ. The factor loadings, standard errors and statistical significance of each factor loading are summarized in Table 2. Based on these results, the indicators were deemed appropriate, and the analysis process proceeded to the next stage.

### 3.3. Measurement Invariance

Invariance testing was conducted to assess whether the same latent construct (stress, DMQ, and FFGC) was being measured across specified groups, in this case, women with HIV and those without HIV. A comparative fit index (CFI) of less than 0.01 indicated that a model had passed the respective test for invariance. Table 3 summarizes the findings from the invariance testing. This model passed all the tests for invariance, with both weak and strong invariance achieving a change in CFI of less than 0.01. 

### 3.4. Multigroup Analysis

Phantom variables were added to the model, and constraints added to test if there were any differences based on HIV status. There was no statistically significant difference between the latent regression model, where the independent variables were allowed to freely estimate between groups, and the regression constrained model (Table 4). This suggests that the ability of the independent variables to predict FFGC was not moderated by HIV status. As such, the strength of correlations among the variables, and their predictive capabilities was the same in both groups, women with and without HIV. 

### 3.5. Structural Equation Model Findings

In Table 5, a 1 kg/m² increase in GBMI was associated with a 0.223 unit increase in FFGC (*p* < 0.001) and a 1 unit increase in income level was associated with a 0.141 unit increase in FFGC in women without HIV (*p* = 0.041). Higher household incomes were associated with higher dietary micronutrient quality (*p* = 0.002), GBMIs (*p* = 0.012) and food security (*p* = 0.001). Higher levels of stress were associated with lower household incomes (*p* = 0.004) and higher levels food insecurity (*p* < 0.001). Social support was also associated with better household food security (*p* = 0.008). There was a marginally significant inverse association between DMQ and food insecurity, with higher DMQ associated with better food security status. The findings of SEM are summarized in Table 5.

### 3.6. Final SEM Diagram

In the final SEM, Figure 3, the non-significant pathways have been represented by a dotted line. This diagram summarizes the information contained in Table 5.

### 3.7. Modification Indices and Goodness-of-Fit Summary

Table 6 summarizes the model’s goodness of fit indices. The chi-square statistic for the model was 171.14 with 145 degrees of freedom. In terms of goodness of fit indices, root mean square error of approximation (RMSEA) was 0.034 (90% CI: 0.000–0.052) which suggested a good model fit and standardized root mean square residual (SRMR) was 0.060, suggesting an acceptable fit. Comparative fit index (CFI) was 0.968 and Tucker-Lewis index (TLI) was 0.959. Since both were above 0.9, this suggests a good model fit. Overall, when considering all modification indices, the model performed well.

## 4. Discussion

This study used a more holistic approach to understand the complex interrelationships between social and nutritional prenatal exposures such as income, food security, stress, micronutrient quality and weight status in this cohort of women. Based on previous work, when looked at separately, all these factors have been associated with reduced length, age, or weight at birth [8,9,10,11,12,13] and using this latent variable, FFGC allowed for a more comprehensive estimation of fetal growth conditions. A 1 unit increase in GBMI was associated with a 0.223 unit increase in FFGC (*p* < 0.001). These findings are in line with several other studies that have shown an association between higher GBMI and birth weights, such as in a relatively recent study in Vietnam [45], in Nepal [46], and in China [47]. The positive associations between GBMI, gestational age at birth and birth weight have also been documented in systematic reviews [48,49]. However, high GBMIs, while protective against low birth weight, do pose other risks such as increased blood pressure and gestational diabetes in the mother, and fetal hypoglycemia (low blood glucose), poor birth outcomes and spontaneous abortion of the fetus [49,50,51] and increased risk of a large for gestational age infant [31,52,53,54,55]. 

In SA, inequality is still a challenge that exposes certain population groups to several factors that can hinder the achievement of FFGC, such as the higher rates of poverty and unemployment [56]. Income levels influence everything from health seeking behaviors to dietary practices [57], therefore it was not surprising that a 1 unit increase in income level was associated with a 0.141 unit increase in FFGC in pregnant women (*p* = 0.041). This is in line with a recent USA study that found an association between income and birth outcomes, where a $10,000 increase in a county’s median income was associated with 0.34 fewer low birth weight cases per 100 live births [58]. Similar positive associations have been observed between income and birth outcomes in Ethiopia, Angola, and China [59,60,61]. 

We examined the associations between dietary micronutrient quality on FFCG and did not find any associations. This was contrary to studies that have found associations between diet quality and FFCG. Analysis of data from over 170,000 women sampled in India’s National Family Health Survey observed that low birth weight was associated with diet quality and iron intake [62]. A possible reason for this is the fact that very few of the women in the current study met the requirements for good diet quality. 

Higher household incomes were associated with higher food security (*p* = 0.001) and higher dietary micronutrient quality (*p* = 0.002). This is to be expected since higher incomes are often associated with food security and better diet quality [63,64]. While we did not see an association between stress and FFCG, higher stress was associated with lower household incomes (*p* = 0.004) and higher levels of food insecurity (*p* < 0.001). This supports the findings from other studies that have observed higher levels of stress and even depression in households with lower food security, which also happened to have lower incomes [65,66,67]. There was also a marginally significant inverse association between DMQ and food insecurity (*p* = 0.06), with higher DMQ associated with better food security status, which is in line with the existing literature that shows associations between food security and diet quality [68,69,70].

### 4.1. Strengths of the Study

One of the major strengths of this study is that it used a latent variable approach to measure FFGC, distinguishing it from most studies that rely on birthweight as the only proxy indicator of the fetal environment. Another strength of the study is that it is the first study that we are aware of, that explores both nutritional and social factors, and considers not only how these affect FFGC, but also how social and nutritional factors are associated with each other. By doing this, we were able to not only acknowledge, but also model, some of the complexity of real-life settings. Our results therefore provide a useful perspective that can add value to the work of practitioners and policymakers alike, by highlighting key issues to prioritize and allowing them to focus resources and effort on the issues most likely to result in the most significant improvements in FFGC. The study included a multigroup analysis, requiring formal tests to determine if the model was valid regardless of HIV status. This is an important perspective, due to the relatively high HIV prevalence in the country. Several measures were put in place to ensure the accuracy and reliability of measurements in the study, such as recruiting skilled fieldworkers and training them to standardize data collection methods. 

### 4.2. Limitations of the Study

The major limitation of this study was the poor completion rate (incomplete birth outcome data) in this study, partly to be expected due to the longitudinal nature of the study, but further exacerbated by the COVID-19 pandemic. Women lost to follow-up did not provide the birth outcome information and HIV status-related information and were therefore excluded from the SEM analysis, which may have impacted the power necessary to detect associations between stress and FFGC, as well as DMQ and FFGC in our sample. Another limitation of this study was the use of self-reported data for the collection of dietary intake information due to recall bias. Since this was a cohort study, purposive sampling was used, which might not provide results that are generalizable to the whole population. Another issue was the fact that the recruitment site was a referral hospital where high risk pregnancies were referred which increases the chances of identifying poor birth outcomes. Lastly, women were recruited at various stages of gestation with no subsequent follow-up until the infant was born.

## 5. Conclusions

Contrary to expectations, DMQ was not associated with FFGC, but GBMI and household income were the only two factors directly related to FFGC. This study provides a more comprehensive estimation of the fetal environment using the latent variable, FFGC. The use of structural equation modeling allows for a more holistic investigation of components that are otherwise difficult to measure or quantify due to their latent nature. The results of this study show that attention must be given to addressing poverty and maternal GBMIs, to optimize the in-utero environment. If time and effort is invested into improving income levels and GBMI of pregnant women, the prevalence of poor birth outcomes in SA might start to decrease. We were able to identify two key issues, and our results may guide not only future research, but also, decision making regarding programs and policies in SA. Prenatal programs that optimize BMI before pregnancy may also be beneficial to the community. We also gained insight into the potential role of social support in buffering against food insecurity, as well as the positive association between income and both food security and dietary quality. Stress was associated with low income and food insecurity, and to reduce the amount of stress experienced during pregnancy, strategies to improve incomes and food security, must be implemented. Overall, a multisectoral approach may be useful in targeting the different nutritional and social issues that contribute to fetal growth conditions.

## Figures and Tables

**Figure 1 nutrients-14-04642-f001:**
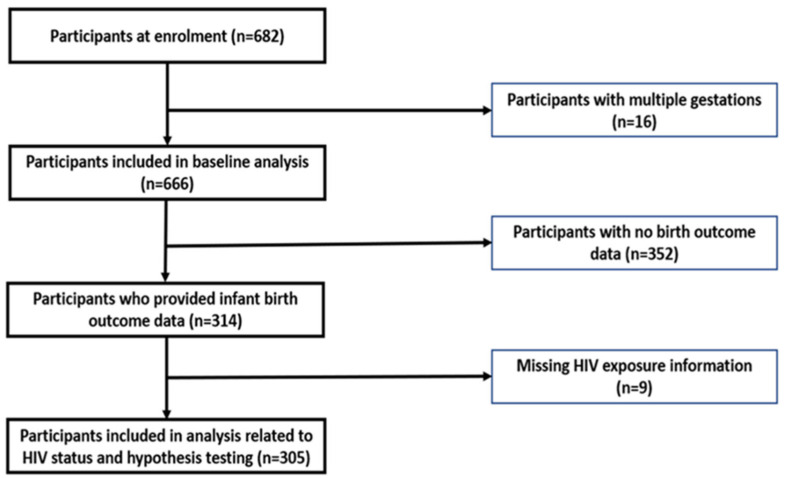
Flowchart of participants throughout the study.

**Figure 2 nutrients-14-04642-f002:**
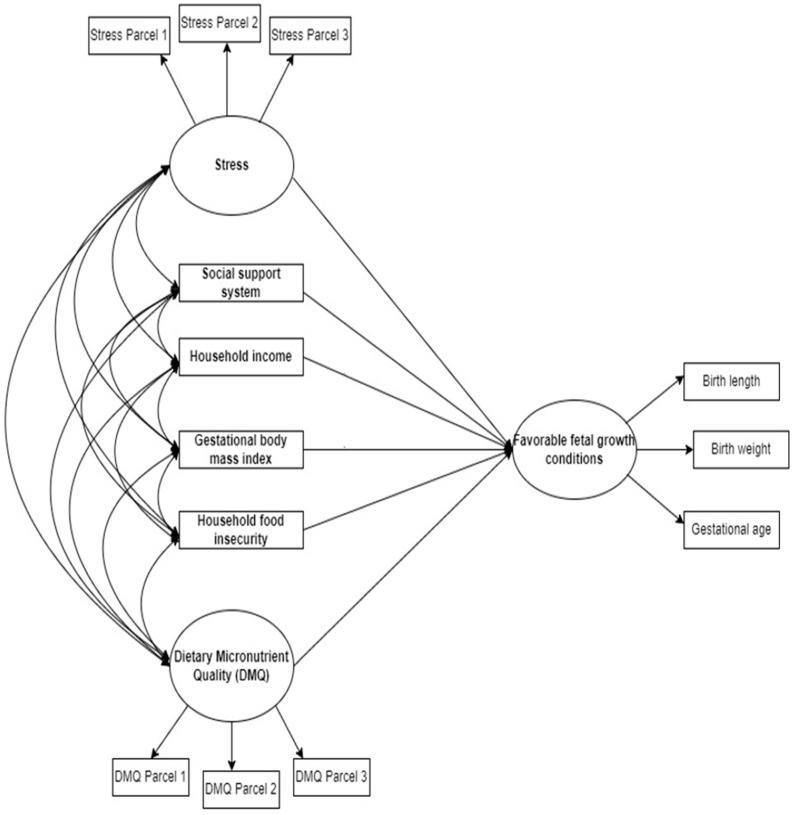
Hypothesized SEM of the social and nutritional factors associated with favorable fetal growth conditions in South African women.

**Figure 3 nutrients-14-04642-f003:**
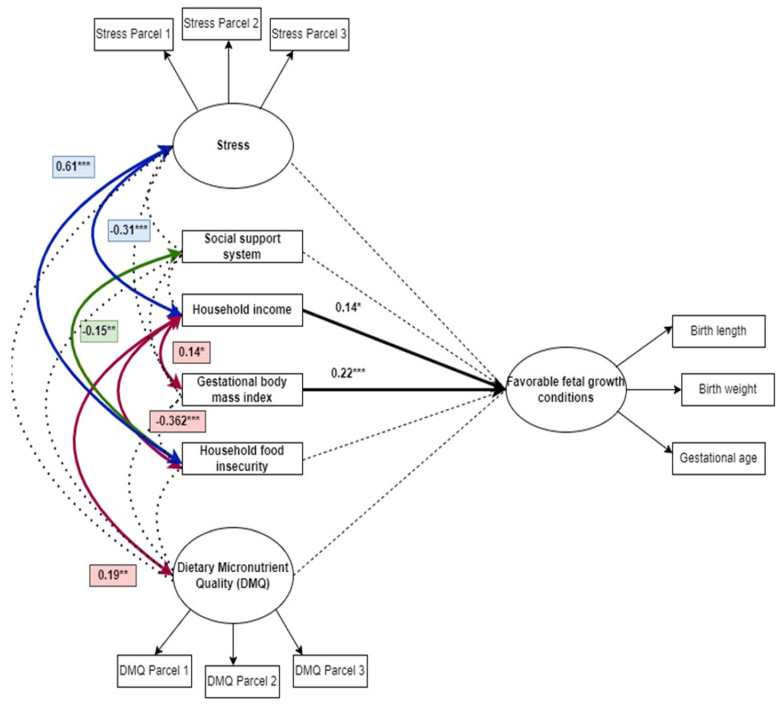
Final model of the nutritional and social determinants of favorable fetal growth conditions. * *p* < 0.05, ** *p* < 0.01, *** *p* < 0.001.

**Table 1 nutrients-14-04642-t001:** Comparison of age and pregnancy stage at enrollment based on HIV status.

	Woman’s Age at Enrolment (Years)			Gestational Age at Enrolment (Weeks)		
	Median	IQR	*p*-Value	Median	IQR	*p*-Value
Women with HIV	34.0	29–37	0.014 *	32.0	25–36	0.13
Women without HIV	32.0	27–37		33.0	27–37	

* *p* < 0.05 considered statistically significant using Kruskal-Wallis test.

**Table 2 nutrients-14-04642-t002:** Relationships between the latent variables and their indicators.

		Group 1—Women Living with HIV			Group 2—Women without HIV		
Latent Variable	Unit	Loading	Standard Error	*p*-Value	Loading	Standard Error	*p*-Value
Dietary micronutrient quality (DMQ)	DMQ Parcel 1	0.661	0.045	<0.001 *	0.745	0.045	<0.001 *
	DMQ Parcel 2	0.709	0.053	<0.001 *	0.837	0.053	<0.001 *
	DMQ Parcel 3	0.760	0.055	<0.001 *	0.771	0.055	<0.001 *
Stress	Stress Parcel 1	0.143	0.140	0.015 *	0.208	0.140	0.015 *
	Stress Parcel 2	0.377	0.292	<0.001 *	0.480	0.292	<0.001 *
	Stress Parcel 3	0.427	0.305	<0.001 *	0.504	0.304	<0.001 *
Favorable fetal growth conditions	Birth weight	0.857	0.049	<0.001 *	0.776	0.049	<0.001 *
	Birth length	0.895	0.053	<0.001 *	0.827	0.053	<0.001 *
	Gestational age	0.695	0.055	<0.001 *	0.613	0.044	<0.001 *

* *p* < 0.05 considered statistically significant.

**Table 3 nutrients-14-04642-t003:** Invariance Testing.

	χ^2^	Df	RMSEA	SRMR	*p*	TLI	CFI	ΔCFI	Pass?
Configural model	125.46	96	0.043	0.054	0.023	0.940	0.963		
Weak invariance	134.15	102	0.044	0.058	0.018	0.938	0.959	0.003	Yes
Strong invariance	139.26	108	0.043	0.060	0.023	0.943	0.960	−0.001	Yes

**Table 4 nutrients-14-04642-t004:** Multigroup analysis.

Model	χ^2^	df	RMSEA	RMSEA 90% CI	SRMR	CFI	TLI	ANOVA	Pass?
Latent regression model	139.26	108	0.043	0.017–0.062	0.060	0.960	0.943		
Regression constrained model	159.57	129	0.039	0.011–0.057	0.067	0.961	0.953	*p* > 0.05	Pass

**Table 5 nutrients-14-04642-t005:** Associations within the structural equation model/Hypothesis Tests.

Path	β (SE)	*p*
Stress and favorable fetal growth conditions (FFGC)	−0.032 (0.169)	0.84
Social support and FFGC	0.022 (0.063)	0.72
Household income and FFGC	0.141 (0.072)	0.041 *
Gestational body mass index (GBMI) and FFGC	0.223 (0.067)	<0.001 *
Household food security and FFGC	0.062 (0.177)	0.58
Dietary micronutrient quality (DMQ) and FFGC	0.075 (0.073)	0.28
Path	r (SE)	*p*
Stress and dietary micronutrient quality	−0.009 (0.114)	0.94
Stress and household income	−0.313 (0.109)	0.004 *
Stress and GBMI	−0.135 (0.097)	0.17
Stress and support system	−0.090 (0.098)	0.36
Stress and household food insecurity	0.605 (0.102)	<0.001 *
DMQ and household income	0.193 (0.061)	0.002 *
DMQ and GBMI	0.061 (0.062)	0.33
DMQ and support system	−0.004 (0.062)	0.95
DMQ and household food insecurity	−0.188 (0.063)	0.061
Household income and GBMI	0.140 (0.055)	0.012 *
Household income and support system	0.028 (0.056)	0.61
Household income and household food insecurity	−0.362 (0.049)	<0.001 *
GBMI and support system	−0.050 (0.056)	0.37
GBMI and household food insecurity	−0.080 (0.056)	0.15
Support system and household food insecurity	−0.145 (0.055)	0.008 *

* *p* < 0.05 considered statistically significant.

**Table 6 nutrients-14-04642-t006:** Goodness of fit summary and interpretation.

Model	χ^2^	df	RMSEA	RMSEA 90% CI	SRMR	CFI	TLI
SEM	159.57	129	0.039	0.011–0.057	0.067	0.961	0.953
Interpretation			Good		Acceptable	Good	Good

## Data Availability

The datasets used and/or analyzed during the current study are available from the corresponding authors on reasonable request.

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
