# Peer review of "Effects of Nutritional and Social Factors on Favorable Fetal Growth Conditions Using Structural Equation Modeling"

_nutrients, 2022, doi:10.3390/nu14214642_

Round 1
Reviewer 1 Report
This manuscript examined the important topic of the effects of nutritional and social factors on favorable fetal 2 growth conditions using structural equation modeling. Overall, the article provides good empirical data, interesting findings.
Author Response
25 October 2022
We want to thank the reviewers for giving their time and expertise to this document. Please find below our responses to your comments describing how we have addressed them in the paper.
|
Reviewer #1: |
|
|
Comment |
Response |
|
This manuscript examined the important topic of the effects of nutritional and social factors on favorable fetal 2 growth conditions using structural equation modeling. Overall, the article provides good empirical data, interesting findings |
Thank you for your review and feedback. |

Reviewer 2 Report
This is an important topic and a refreshing, novel, holistic and valuable approach. The lack of influence of HIV infection is especially interesting, suggesting that effects are mostly mediated through influences on the other domains via income and BMI. Likewise the lack of direct effects of stress and micronutrients.
Minor suggestions are:
It would be helpful to readers to explain the rationale for parcelling variables for stress and DMQ in this analysis, also what was being averaged in each parcel and whether any transformations were done.
In figure 1 the arrows from the left hand side leading to FFGC suggest a formative rather than a reflective relationship. It may be helpful to confirm the specificaiton and explain to readers since reflective models are more common and there are differences in treatment of covariance.
Despite reference 43, I suspect a larger sample size may have provided additional information on stress and DMQ and should be regarded as a potential limitation rather than beinf described as adequate.
Author Response
25 October 2022
We want to thank the reviewers for giving their time and expertise to this document. Please find below our responses to your comments describing how we have addressed them in the paper.
|
Reviewer #2: |
|
|
Comment |
Response |
|
It would be helpful to readers to explain the rationale for parceling variables for stress and DMQ in this analysis, also what was being averaged in each parcel and whether any transformations were done. |
I have added the following statement to explain the rationale for parceling the variables in Line 160-166: At times a manifest variable may only tell a part of the story, thus combining several variables into a parcel creates a more reliable indicator. Parceling is a technique that improves the quality of the indicator, and averaging, as opposed to summing the items allows the parcels to have similar metrics [43]. Based on the balancing approach, items with the strongest item–scale correlations were paired with those with the weakest, and this process was repeated until all items were parceled. |
|
In figure 1 the arrows from the left hand side leading to FFGC suggest a formative rather than a reflective relationship. It may be helpful to confirm the specification and explain to readers since reflective models are more common and there are differences in treatment of covariance. |
The arrows pointing towards FFGC are the hypothesized regression associations. The FFGC indicator is reflective too, and its indicators are birth length, birth weight and gestational age.
I have summarized this in lines 189-194 and added an extra sentence to hopefully make this clearer. |
|
Despite reference 43, I suspect a larger sample size may have provided additional information on stress and DMQ and should be regarded as a potential limitation rather than being described as adequate. |
Line 336-339 has been rephrased as follows: Women lost to follow-up did not provide the birth outcome information and HIV status-related information and were therefore excluded from the SEM analysis, which may have impacted the power necessary to detect associations between stress and FFGC, as well as DMQ and FFGC in our sample. |
We hope you will find this in order.
